# Talking to GDELT Through Knowledge Graphs

**Audun Myers, Max Vargas, Sinan G. Aksoy, Cliff Joslyn, Benjamin Wilson, Lee Burke, Tom Grimes**

**Editors:** Leilani H. Gilpin, Eleonora Giunchiglia, Pascal Hitzler, and Emile van Krieken

## Abstract

In this work we study various Retrieval Augmented Generation (RAG) approaches to gain an understanding of the strengths and weaknesses of each approach in a question-answering analysis. To gain this understanding we use a case-study subset of the Global Database of Events, Language, and Tone (GDELT) dataset as well as a corpus of raw text scraped from the online news articles. To retrieve information from the text corpus we implement a traditional vector store RAG as well as state-of-the-art large language model (LLM) based approaches for automatically constructing KGs and retrieving the relevant subgraphs. In addition to these corpus approaches, we develop a novel ontology-based framework for constructing knowledge graphs (KGs) from GDELT directly which leverages the underlying schema of GDELT to create structured representations of global events. For retrieving relevant information from the ontology-based KGs we implement both direct graph queries and state-of-the-art graph retrieval approaches. We compare the performance of each method in a question-answering task. We find that while our ontology-based KGs are valuable for question-answering, automated extraction of the relevant subgraphs is challenging. Conversely, LLM-generated KGs, while capturing event summaries, often lack consistency and interpretability. Our findings suggest benefits of a synergistic approach between ontology and LLM-based KG construction, with proposed avenues toward that end.

## 1. Introduction

This work studies Retrieval Augmented Generation (RAG) for corpus communication and analysis using Large Language Models (LLMs). Our goal is to understand RAG strategy benefits and drawbacks with LLMs, using a novel Knowledge Graph (KG) derived from the Global Data on Events, Location, and Tone (GDELT)[1] Leetaru and Schrodt (2013) dataset as a case study.

To enhance LLM outputs, researchers have applied LLMs to query proprietary data via RAG Lewis et al. (2020). However, reasoning over typical RAG frameworks, which rely solely on unstructured text, often fails to capture global information Edge et al. (2024); Xu et al. (2024). Motivated by this, interest has grown in adapting these techniques for graph-structured data, enabling LLMs to directly ingest key knowledge base relationships Edge et al. (2024); He et al. (2024); Mavromatis and Karypis (2024); Zhu et al. (2023). KGs Hogan et al. (2021) are richly attributed graph structures with semantic information on nodes and edges. KG techniques facilitate automatic information extraction and querying without explicit knowledge of query languages, typically by finding subgraphs that answer user queries.

---

1. https://www.gdeltproject.org/

The interactions between KGs and LLMs extend beyond QA and knowledge extraction Pan et al. (2024). While KGs enhance LLM outputs, LLMs can also enhance existing KGs or create new ones. However, existing techniques often either (1) do not impose different ontological structures during graph creation or (2) only focus on extracting ontological structures using LLMs Trajanoska et al. (2023); Yao et al. (2024).

Throughout this work, we use the GDELT dataset as a case study. GDELT is a massive collection of news reports, updated every 15 minutes, providing a real-time computational record of global events. It aggregates information from diverse sources, including people, organizations, locations, themes, and emotions. GDELT offers a snapshot of global events, enabling researchers to explore complex patterns, identify trends, assess risks, understand public sentiment, and track issue evolution. Its applications are diverse, including event monitoring Owuor and Hochmair (2023); Owuor et al. (2020); Yonamine (2013a), risk assessment and prediction Galla and Burke (2018); Qiao and Chen (2016); Qiao et al. (2017); Voukelatou et al. (2020); Wu and Gerber (2017); Yonamine (2013b), and social science research Alamro et al. (2019); Bodas-Sagi and Labeaga (2016); Boudemagh and Moise (2017); Keertipati et al. (2014).

GDELT describes its structure as a Global Knowledge Graph (GKG, specifically GKG2). In reality, GDELT-GKG2 is implemented as multiple linked tables, effectively a relational database. Another important contribution is realizing GKG2 as a proper KG (a graph database), derived from and consistent with its native relational database form. To facilitate this, we identified a lightweight ontology for GDELT, reflecting its relational database schema in a KG form. Using our constructed GDELT-GKG2 KG, we explore LLM-based tools for information extraction and confirm its utility for question-answering where traditional RAG fails. Our analysis includes a comparison to KGs produced by processing news articles with an LLM, prompting it to adhere to a reduced version of the same ontology.

The current neurosymbolic landscape features numerous experimental architectures. Details are in Section 3.2, but we preview our five quantitatively compared methodological pathways in Figure 1: 1) graph queries on the DKG (derived "directly" from GKG2); 2) G-Retriever[2] He et al. (2024) against the same DKG; 3) RAG against a vector store of GKG2; 4) G-Retriever against the LKG (derived using Llamaindex[3] LLa on GDELT source articles); and 5) GraphRAG[4] Q&A against the GRKG, using Microsoft's open-source GraphRAG package with default settings.

## 2. Constructing a Knowledge Graph for GDELT

As previously mentioned, while the GDELT-GKG2 dataset is not natively in the form of a knowledge graph, it is advertised and frequently cited as being one. Our distinct contribution is converting this popular relational database into a proper KG.

GKG2 natively comprises three related tables: 'expert.csv' for event information, 'GKG.csv' for article data, and 'mentions.csv' to link articles and events.

The database schema for these three CSV files is shown in Fig. 2 (see also Jayanetti et al. (2023)). The key characteristics of this relational schema are:

---

2. https://github.com/XiaoxinHe/G-Retriever

3. https://www.llamaindex.ai/

4. https://microsoft.github.io/graphrag/

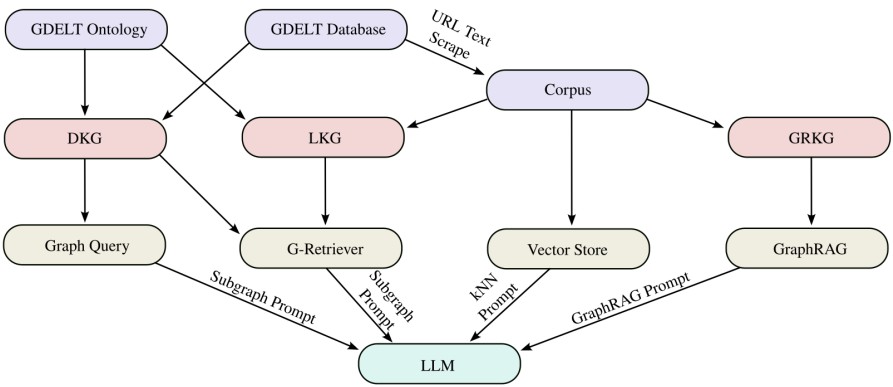

Figure 1: Pipeline of different experiments ran to analyze the GDELT database using an LLM.

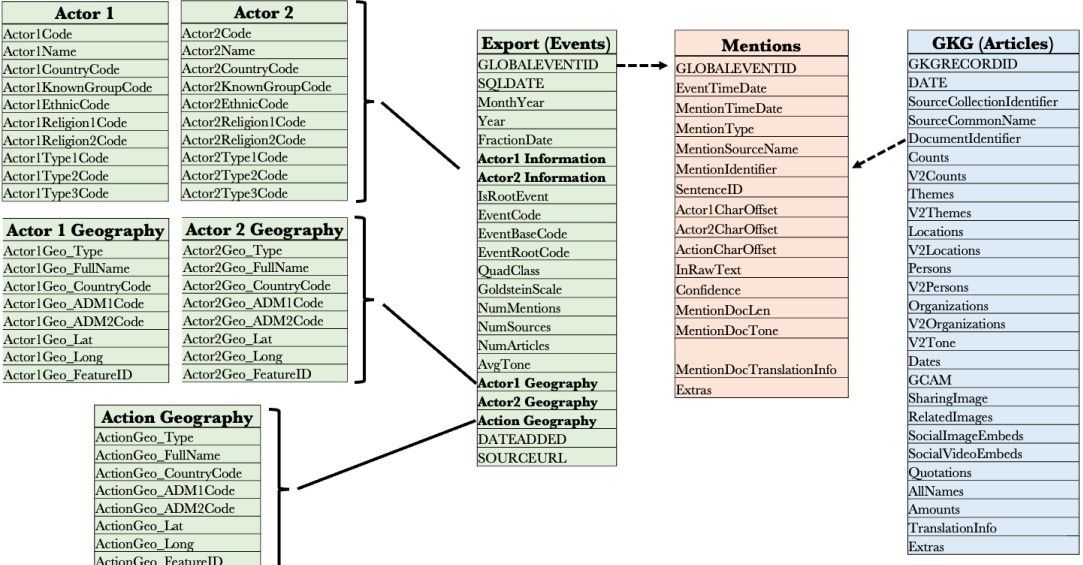

Figure 2: GDELT GKG 2.0 schema relating articles (GKG), mentions, and events (Export).

- Tables are color-coded: Events (green), Mentions (pink), Articles (blue); Events table is split for clarity.

- One-to-many relationships shown by single-headed arrows:

  - Events map to multiple Mentions via 'GLOBALEVENTID'.
  - Articles map to multiple Mentions via 'DocumentIdentifier' (Article) and 'MentionIdentifier' (Mention).

- The Mentions table functions as a many-to-many link between Events and Articles. Each article has unique identifiers ('GKGRECORDID' or 'DocumentIdentifier').

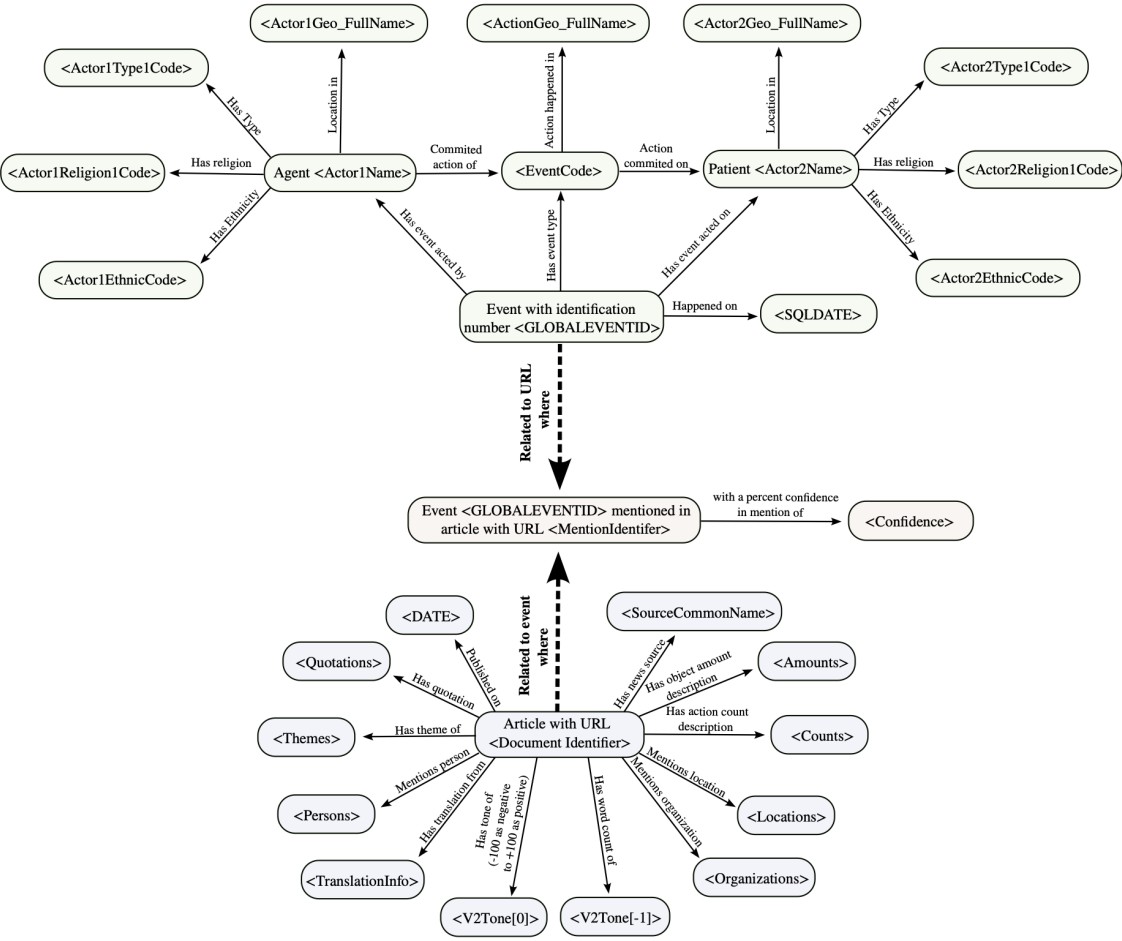

Figure 3: GDELT GKG 2.0 ontology relating articles and events.

Automated methods for deriving graphical forms from relational databases are well-established Sequeda et al. (2011). Typically, table rows become nodes linked to their column values, forming bipartite "star" graphs which connect via shared values. This method generates a graph schema consistent with the RDB, varying in complexity. For GDELT, this approach is largely applicable, though event table constraints necessitate a more specific structure.

After understanding the GDELT database schema, we developed a capability to convert (portions of) the GDELT database to a KG using an ontology as a graph typing schema, derived from the above relational schema. This is shown in Fig. 3, to be interpreted as follows:

- Nodes indicate possible KG node types.

- Nodes are color-coded by source relational table.

- Fields in ⟨angled brackets⟩ denote schema field names.

- Solid edges represent relational table fields, labeled by semantic relation type.

- Dashed and bold edges indicate structural one-to-many relational schema relations.

The naming convention ensures unique identifiers: ⟨GLOBALEVENTID⟩ for Events, ⟨DocumentIdentifier⟩ for Articles, and (⟨GLOBALEVENTID⟩ , ⟨MentionIdentifier⟩) for Mentions. Note that 'DocumentIdentifier' and 'MentionIdentifier' refer to the same field, often a URL, but have different names.

## 3. Case Study - Baltimore Bridge Collapse

This section analyzes question-answering (QA) data from the March 26, 2024, Francis Scott Key Bridge collapse in Baltimore. Recent data was chosen as LLMs were not yet trained on these events, necessitating knowledge systems. We analyzed a GDELT subset (12:00 AM - 10:00 AM EST), capturing initial media response. Data filtered by "Baltimore," "bridge," "collapse," or "ship" keywords comprised 1.33% of available data: 371 events, 2047 mentions, and 209 articles.

### 3.1. GDELT Knowledge Graphs

Three KGs were constructed from GDELT data and scraped text:

**Direct KG (DKG):** A direct conversion of the GDELT subset into a KG based on our ontology (Fig. 3, Fig. 3.1).

**LlamaIndex KG (LKG):** Generated by an LLM (Mixtral-8x7B Jiang et al. (2024)) from scraped source articles (209 URLs), incorporating ontology knowledge LLa (Fig. 3.1).

**GraphRAG KG (GRKG):** Created from the same articles as LKG, using Microsoft's open-source GraphRAG package with default parameters (Fig. 3.1).

The example DKG (Fig. 3.1) has 3,469 nodes and 18,052 edges, with nodes color-coded by source and labels omitted for clarity.

LKG construction (Fig. 3.1) used Mixtral-8x7B Jiang et al. (2024) and LlamaIndex's procedure LLa. The default prompt, without ontology, resulted in a star graph. Even a modified prompt with the full ontology yielded a star graph. Nontrivial graph structure only appeared when prompting with a reduced ontology adapted for unstructured text, requesting specific vertex types ("Event", "Article", "Mention", "Person", "Quotation", "Organization", "Location", "Other") and edge types (e.g., "Related to event where"). While this yielded a non-trivial KG, adherence to prescribed node/edge types was challenging, possibly due to hallucinations. This pipeline also had entity/relation resolution issues (e.g., 'Container ship' vs. 'Container_ship.').

GRKG construction used Llama-3.1-8B Team (2024) due to Mixtral-8x7B's context window limitations. GraphRAG indiscriminately generates relations, identifying default

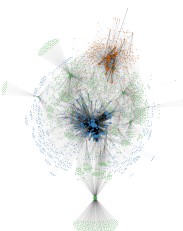
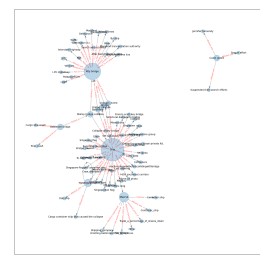
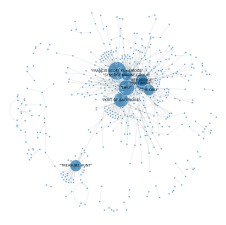

(a) Example DKG constructed from ontology with no labels, but color coding set to match ontology.

(b) Example LKG constructed corpus of text using LlamaIndex.

(c) Example GRKG constructed corpus of text using GraphRAG, removing all isolated nodes. Large nodes have degree $\geq 25$.

Figure 4: KG formations from GDELT Data of Baltimore Bridge collapse event. This subset of data included 27 articles with 283 related mentions to 143 events during the Baltimore bridge collapse from midnight to 10:00 AM EST. The corpus of text was created from web scraping the 27 URLs associated to the articles.

entities: "organizations", "persons", "geo(locations)", and "events." Like LlamaIndex, GraphRAG struggled with entity resolution (e.g., "DALI" vs. "THE DALI"), producing many small, often isolated, components (435 of 968 nodes were isolated).

## 3.2. Knowledge Graph Analysis Methodology

Fig. 4 highlights the size differences among DKG, LKG, and GRKG, likely due to LlamaIndex and GraphRAG's summarizing nature. The LKG's LLM struggled with ontology adherence, generating unprescribed but interpretable edge types (e.g., '(Maersk, Chartered, Container ship)').

To validate our ontology and demonstrate its uses, we qualitatively compared LLM-based QA on the GDELT dataset. Fig. 1 shows our five LLM pipelines for querying GDELT, using 7-8B parameter models for final processing (Llama-3-8B for GraphRAG; Mistral-7B with E5-large-v2 embeddings for others). Pipelines, left to right:

1. Direct graph queries to DKG: Analyst converts natural language to graph queries; LLM interprets/repackages results.

2. G-retriever on DKG: Automatically fetches DKG subgraph from natural language; requires vectorizing KG nodes/edges; LLM interprets subgraph.

3. LlamaIndex KG (LKG) from full-text documents: KG built by LLM parsing scraped GDELT URLs; proceeds as in (2).

4. Vector Store RAG: Vectorizes articles (500-token chunks) via embedding model; given question, extracts similar snippets using Euclidean metric; LLM processes question and context.

5. GraphRAG QA on GRKG: KG built using GraphRAG ecosystem; utilizes its QA capabilities.

LLM use in method (1) is often unnecessary as answers can be inferred directly from graph query output; it primarily repackages results. Method (1) serves as a DKG 'ground-truth' baseline: a suitable query confirms answer presence or absence.

Graph queries are applied only to DKG, not LLM-produced KGs (LlamaIndex or GraphRAG), due to their less defined structure complicating useful query formulation.

### 3.3. Results

Table 1 presents sample questions evaluated across the five pipelines (Fig. 1). Exact GDELT KG queries used keywords (e.g., "Bridge, Collapse, River"; "CNN, Baltimore, Bridge, Collapse"; "Brandon Scott") to search edge triples (converted to sentences) and extract edge-induced subgraphs. The prompt was: "Please answer the question given the following information:" appended with edge sentences.

- What is the name of the Bridge that collapsed and what river was it on?: **Bridge, Collapse, River**

- What is the name of the ship that collided with the baltimore bridge?: **Ship, Collide, Baltimore, Bridge**

- How many articles did CNN publish about the baltimore bridge collapse?: **CNN, Baltimore, Bridge, Collapse**

- On what date did the Baltimore Bridge collapse?: **Date, Baltimore, Bridge, Collapse**

- Who is Brandon Scott?: **Brandon Scott**

- Who is Niki Fennoy?: **Niki Fennoy**

- What are the top themes present in the data?: **Has_Theme**

Vector store, GraphRAG, and DKG direct queries yielded the best QA results. Direct DKG queries excelled at high-level/aggregate questions (e.g., recurrent themes, news sources). GraphRAG supported high-level questions but performed stronger on fine-grained ones, similar to vector stores. G-retriever's automated search often failed to retrieve accurate subgraphs, especially with LlamaIndex KGs; fine-tuning might improve this. Nevertheless, direct data probing via hand-crafted queries remains valuable, indicating automated information extraction needs further development to match baseline corpus-wide reasoning.

Combining DKG direct query results with GraphRAG and standard RAG produced suitable answers for all questions. DKG provides better responses for high-level/aggregate data, while vector store and GRKG identify local document information potentially missing in DKG.

Direct LLM prompting (without KGs or vector stores) independently identified Brandon Scott as Baltimore's mayor. However, due to recency, the LLM could not answer other posed questions independently.

For quantitative evaluation, Table 2 provides manually curated ground truth answers as benchmarks.

Table 1 displays sample questions evaluated across the five pipelines (Fig. 1). Exact GDELT KG queries used keywords (e.g., "Bridge, Collapse, River"; "CNN, Baltimore, Bridge, Collapse"; "Brandon Scott") to search edge triples (converted to sentences) and

| | Graph Query on DKG | G-Retriever on DKG | RAG using Vector Store | G-Retriever on LKG | GraphRAG Q&A on GRKG |
|---|---|---|---|---|---|
| What is the name of the Bridge that collapsed and what river was it on? | The Francis Scott Key Bridge and it was on the Patapsco River in Maryland. | The bridge is located in Sri Lanka. However, there is no explicit mention of the river's name. | The Francis Scott Key Bridge collapsed into the Patapsco River. | The bridge that collapsed spanned over the Patapsco river. | The Francis Scott Key Bridge which spans the the Patapsco River. |
| What is the name of the ship that collided with the baltimore bridge? | The name of the ship is not provided. However, the ship is a container ship and Singapore-flagged. | There is no mention of the name of the ship that collided with the bridge. | The name of the ship that collided with the Baltimore bridge is the Dali. | The name of the ship that collided with the Baltimore bridge is Maersk. | The ship that collided with the Baltimore bridge is the Dali. |
| How many articles did CNN publish about the Baltimore bridge collapse? | CNN published 5 articles about the Baltimore bridge collapse. | CNN published a total of 2 articles about the Baltimore bridge collapse. | CNN published three articles about the Baltimore bridge collapse. | There is only one article published by CNN about the Baltimore bridge collapse. | CNN published at least two articles about the Baltimore bridge collapse. |
| On what date did the Baltimore Bridge collapse? | The Baltimore Bridge collapsed on March 26, 2024. | I cannot directly answer that question based on the given data. | The Baltimore Bridge collapsed on March 26, 2024. | The Baltimore Bridge collapsed at 1:20 a.m. | The Baltimore Bridge collapsed on March 26, 2024. |
| Who is Brandon Scott? | Brandon Scott is a person mentioned in several news articles related to the collapse of the Francis Scott Key Bridge. | Brandon Scott is the Mayor of Baltimore. | Brandon Scott is the mayor of Baltimore, Maryland. | Brandon Scott is not mentioned in the given data. | Brandon Scott is the mayor of Baltimore, Maryland. |
| Who is Niki Fennoy? | Niki Fennoy is a person mentioned in various news articles related to the collapse of the Francis Scott Key Bridge. | The article from `thepeninsular-qatar.com` mentions Niki Fennoy. | I don't know. Niki Fennoy is not mentioned in the provided context. | Niki Fennoy is not present in the given data. | Niki Fennoy is a city police spokesman. |
| What are the top themes present in the data? | MARITIME_INCIDENT MARITIME MANMADE_DISASTER TAX_FNCACT WB_137_WATER. | MARITIME_INCIDENT CRISIS TAX NEWS ETHNICITY. | I don't have enough information from to determine specific the themes present in the data. | EVENTS AND THEIR RELATIONSHIPS, LOCATIONS, ORGANIZATIONS, VESSELS. | NEWS AND UPDATES BRIDGE COLLAPSE CONSTRUCTION CREW SEARCH AND RESCUE COMMUNITY REPORT. |

Table 1: Table of example questions and answers highlighting deficiencies in each method for analyzing the GDELT data. Table highlight color legend: Green is a correct answer, yellow is a partially correct answer, red is an incorrect answer, and grey is for no answer provided.

extract edge-induced subgraphs. The prompt was "Please answer the question given the following information:", appended with edge sentences.

Vector store, GraphRAG, and DKG direct queries yielded the best QA results. Direct DKG queries excelled at high-level/aggregate questions (e.g., recurrent themes, news sources). GraphRAG supported high-level questions but performed stronger on fine-grained ones, similar to vector stores. G-retriever's automated search often failed to retrieve accurate subgraphs, especially with LlamaIndex KGs; fine-tuning might improve this. Nevertheless, direct data probing via hand-crafted queries remains valuable, indicating automated information extraction needs further development to match baseline corpus-wide reasoning.

Combining DKG direct query results with GraphRAG and standard RAG produced suitable answers for all questions. DKG provides better responses for high-level/aggregate data, while vector store and GRKG identify local document information potentially missing in DKG.

Direct LLM prompting (without KGs or vector stores) independently identified Brandon Scott as Baltimore's mayor. However, due to recency, the LLM could not answer other posed questions independently.

For quantitative evaluation, Table 2 provides manually curated ground truth answers as benchmarks.

| Question | Ground Truth |
| --- | --- |
| What is the name of the Bridge that collapsed and what river was it on? | The Francis Scott Key Bridge on the Patapsco River. |
| What is the name of the ship that collided with the baltimore bridge? | The ship was named the Dali. |
| How many articles did CNN publish about the Baltimore bridge collapse? | CNN published 5 articles. |
| On what date did the Baltimore Bridge collapse? | The collapse occurred on March 26, 2024. |
| Who is Brandon Scott? | Brandon Scott is the Mayor of Baltimore. |
| Who is Niki Fennoy? | Niki Fennoy is a city police spokesman. |
| What are the top themes present in the data? | Themes include maritime incidents, manmade disaster, and water-related topics. |

Table 2: Ground Truth Answers for the Baltimore Bridge Collapse Questions

Figure 5 quantitatively validates qualitative findings by comparing semantic similarity of answers generated by the five methods for the Baltimore bridge collapse. Cosine similarity (calculated using the `sentence-transformers/all-MiniLM-L6-v2` model from the Sentence Transformers library) measures predicted-to-ground-truth similarity; higher values indicate greater accuracy. Box plots show similarity score distributions. Consistent with qualitative analysis, direct graph queries on DKG, standard RAG with vector store, and GraphRAG Q&A on GRKG generally achieve the highest cosine similarity, confirming superior performance over G-Retriever, especially with LKG. This visualization quantitatively supports trends from Table 1.

## 4. Conclusion

This work introduced an ontology to create a richly structured knowledge graph from the GDELT GKG database. While the resulting large KG offers rich information for question-answering, its *ad hoc* graph exploration techniques need further investigation for reliable use. The KG's information is not flawless, yet this case study should prove useful for the broader synergy between KGs and LLMs. LlamaIndex KGs summarized Baltimore bridge collapse events but lacked structural quality for question-answering. GraphRAG techniques performed considerably better, though improvements are still needed for global questions, duplicate entity resolution, and ontologically-guided relation extraction. We believe integrating the ontology into relation extraction will enhance tools like GraphRAG, enabling them to better answer quantitative questions previously only addressable by the DKG pipeline (e.g., article count on a topic).

Large language models continue to adapt across domains, and this KG case study presents significant future development opportunities. The debate between using raw documents or a curated KG should be 'integration', not 'either/or'. Our findings demonstrate the significant value of combining both approaches. We hypothesize an integrated system—combining news articles in a vector store with GDELT KG's rich ontological structure via an LLM interface—would offer superior knowledge retrieval. Specifically, the direct knowledge graph

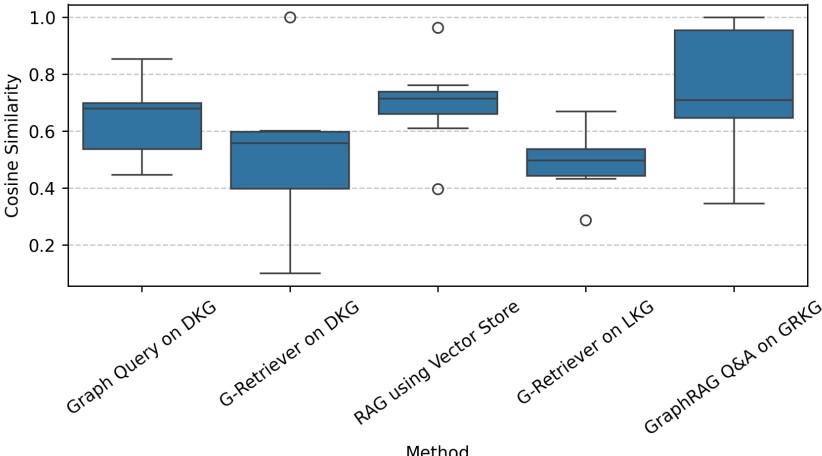

Figure 5: Box plots comparing the cosine similarity scores of different question-answering methods applied to the GDELT data concerning the Baltimore bridge collapse. Higher cosine similarity indicates a greater semantic similarity between the predicted and actual answers.

(DKG) excels at high-level or aggregate questions, providing a strong structured foundation. While the vector store identifies local document information often missing or less accessible in the DKG, GraphRAG has shown effectiveness for both local and global information. Thus, the optimal approach is an integrated system, leveraging DKG for broad context, and the vector store and GraphRAG for detailed, document-specific insights and enhanced global information retrieval. Further work is needed to determine how textual article data can refine this KG; some examples showed DKG information inaccurately reflected article content. Conversely, we must determine how the constructed KG can better search the associated vector store. Research directions include:

- Using LLMs to add new information (entities or relations) to an existing KG. Based on LlamaIndex and GraphRAG observations, careful monitoring is needed to ensure LLM responses adhere to ontological and existing KG structures. Adapting DKG triples to fine-tune the LLM or guide its output with in-context learning could be beneficial.

- Introducing RAG capabilities to fact-check the KG against raw textual information. For example, Niki Fennoy was mentioned in 3 articles but misattributed to 11 others. LLMs offer a potential avenue to fact-check existing relations.

## 5. Acknowledgements

This work is under information release number PNNL-SA-209193.

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
