# OpenReview forum: "Talking to GDELT Through Knowledge Graphs"
_nesyconf.org/NeSy/2025/Conference_Phase_2 — NeSy 2025 - Phase 2 Poster_

### Official Review · Reviewer_ACfz · 2025-07-04
**review of Submission24**

**Rating:** 6
**Confidence:** 4

**Review:**

## summary of the paper

The paper compares the answers generated by multiple different LLM-with-RAG (retrieval augmented generation) pipelines to a common set of questions. Evaluation of the answers is both qualitative (correct, partially correct, incorrect, no answer given) and quantitative. The questions pertain to the bridge collapse event in Baltimore, Maryland on 26 March, 2024 --- an event that occurred 'after' the LLMs used in the Q&A exercise were trained, giving rise to the need for RAG techniques to get useful answers.

The different LLM-with-RAG pipelines retrieve their prompt-augmenting knowledge from one of 3 different knowledge graphs (KGs) that were constructed, using different tools/techniques, from one of 2 different, but related, sets of data concerning the Baltimore bridge collapse.
One of these two sets of data is the GDELT dataset --- an open database of human society that monitor's the world's broadcast, print and web-based news media. GDELT provides a structured distillation of events and news articles. It has a linked-table (relational database) structure:  EVENTS --> MENTIONS <-- ARTICLES. The other set of data is a corpus of the raw text of the news articles scraped from the URLs referenced in the GDELT data --- the source news articles from which the GDELT data would have been distilled.

The GDELT data is used to construct one of the 3 KGs, following an ontology for GDELT introduced by the authors. The text corpus is used to construct the other two KGs. The text corpus is also used to construct a vector store for accessing the corpus directly as part of a traditional RAG pipeline, thus bypassing the use of KGs.

Five different LLM-with-RAG pipelines are evaluated on the common set of questions. The main findings are:
* a) there is no single KG-based LLM-with-RAG pipeline that performs best on all questions, but three of the five perform better than the other two
* b) the KG constructed by restructuring the tabular GDELT data (per the GDELT ontology) helps to answer higher-level questions that require aggregation across the multiple news articles (what the authors call 'global' questions), whereas the KGs constructed from the raw text corpus are sufficient for answering granular questions about specific facts (what the authors call 'local' questions)
* c) combining vector store and KG-based RAG techniques may boost LLM performance and be worth exploring
* d) combining local and global techniques may boost LLM performance and be worth exploring


## evaluation of the paper

**clarity / use of language / quality of writing**

The paper's use of language is OK and it is reasonably clear given the ambitious and complex set of investigations it describes. But I was a reviewer of the 1st submission of this paper, so this is my second encounter with it. So my sense of its clarity may no longer be representative.  I note that the authors acted on my suggestion and shifted their diagram of the 5 LLM-with-RAG pipelines forward in the paper so the reader gets to benefit from it sooner. This works for me, but as I've just explained, my perception of the paper's clarity may no longer be reliable.


**novelty / originality**

The paper's novelty / originality is modest. The authors are primarily exercising and comparing KG-based LLM-with-RAG tools and techniques developed by others. The novel aspect of the research appears to be centred on the conversion of the tabular GDELT data to a KG (collection of triples) representation, according to the GDELT ontology proposed by the authors. This in turn enables the construction of a novel (albeit partly manual and ad hoc) LLM-with-RAG pipeline that leverages this particular KG.

The authors appear to have responded to comments in my review of the 1st submission about there being a known scheme for mapping relational (tabular) data to KG triples. The authors now discuss this scheme, cite a reference, and acknowledge that they rely upon it primarily, but suggest (I think) not exclusively. These changes strengthen the GDELT ontology's credibility, in my view, without diminishing its novelty.

**impact / significance**

The paper's impact / significance is modest. The paper's findings, summarised above, are one indication of this.  Another factor relevant here is that the authors use *ad hoc* techniques for querying the KG converted from GDELT data. They hand-craft their KG queries and run them manually, later feeding the results into an LLM as prompt-augmenting context. The techniques they describe here enable a theoretical investigation that leads to finding (b), but it's not the kind of thing that others would seek to replicate.  More work is needed to automate this LLM-with-RAG pipeline to make it viable in practice.

The authors have supplemented their 1st submission with some quantitative analysis based on measuring the cosine similarity of embeddings of predicted answers and (proposed) ground-truth answers. But the quantitative findings largely echo the qualitative findings and don't change the dial on the paper's conclusions.

**minor observations**

Section 2, page 4, in the paragraph directly below Figure 3:
* The 2nd last sentence of the paragraph ends with the clause "which may or may not be of sufficient complexity to warrant the lofty description 'ontology'". This clause is not helpful to the paper. An ontology needn't be complex to be an ontology, and the adjective 'lofty' feels self-conscious and inappropriate in a scientific paper. Consider removing this clause altogether. The first clause forms a good sentence all by itself.

**Anonymity:**

Remain anonymous

---

### Official Review · Reviewer_4bY8 · 2025-07-07
**Good analytical framework but not evaluated enough**

**Rating:** 3
**Confidence:** 5

**Review:**

My main concern with this paper remains the same and authors, the method has not been evaluated enough to lead to robust conclusions. Authors have just tested the RAG methods on a use case  consisting of only one query about a very specific topic. This use case involves anlysing around 209 articles. Probably this information can be now included in the context window of an LLM without the need of a RAG system. Despite including Fig 5, the analysis is still qualitative and very limited.

**Anonymity:**

Remain anonymous